# CELF Family Proteins in Cancer: Highlights on the RNA-Binding Protein/Noncoding RNA Regulatory Axis

**DOI:** 10.3390/ijms222011056

**Published:** 2021-10-14

**Authors:** Maryam Nasiri-Aghdam, Texali C. Garcia-Garduño, Luis Felipe Jave-Suárez

**Affiliations:** 1División de Inmunología, Centro de Investigación Biomédica de Occidente, Instituto Mexicano del Seguro Social, Guadalajara 44340, Mexico; maryamnasiri.bio@gmail.com; 2Doctorado en Genética Humana, Departamento de Biología Molecular y Genómica, Universidad de Guadalajara, Guadalajara 44340, Mexico; garciatexali@gmail.com; 3Centro Universitario de Ciencias de la Salud, Instituto de Investigación en Ciencias Biomédicas, Universidad de Guadalajara, Guadalajara 44340, Mexico

**Keywords:** CELF proteins, RNA-binding protein, noncoding RNA, cancer

## Abstract

Post-transcriptional modifications to coding and non-coding RNAs are unquestionably a pivotal way in which human mRNA and protein diversity can influence the different phases of a transcript’s life cycle. CELF (CUGBP Elav-like family) proteins are RBPs (RNA-binding proteins) with pleiotropic capabilities in RNA processing. Their responsibilities extend from alternative splicing and transcript editing in the nucleus to mRNA stability, and translation into the cytoplasm. In this way, CELF family members have been connected to global alterations in cancer proliferation and invasion, leading to their identification as potential tumor suppressors or even oncogenes. Notably, genetic variants, alternative splicing, phosphorylation, acetylation, subcellular distribution, competition with other RBPs, and ultimately lncRNAs, miRNAs, and circRNAs all impact CELF regulation. Discoveries have emerged about the control of CELF functions, particularly via noncoding RNAs, and CELF proteins have been identified as competing, antagonizing, and regulating agents of noncoding RNA biogenesis. On the other hand, CELFs are an intriguing example through which to broaden our understanding of the RBP/noncoding RNA regulatory axis. Balancing these complex pathways in cancer is undeniably pivotal and deserves further research. This review outlines some mechanisms of CELF protein regulation and their functional consequences in cancer physiology.

## 1. Introduction

RNA-binding proteins (RBPs) orchestrate crucial aspects of RNA biology in many sets of targets. Global post-transcriptional modifications through RBPs, which are present in almost all organelles of a cell, can exert rapid cellular effects in response to external or internal stimuli. The widespread functions of RBPs range from splicing, localization, modification, stability, and translation of coding and noncoding RNAs [1]. In human cells, it is expected that approximately 1500 RBPs associate with over 33 million interactions in 3′UTR binding sites [2]. RBPs’ multifunctionality, their broad number of targets and subcellular location, as well as their tight interactions with other RBPs and noncoding RNAs, make them a challenging field to research, especially in cancer, where modifications in the expression, mutations, and copy number variations of RBPs can have a significant impact on their performance [3,4].

The CUGBP, ELAV-like family (CELF) of proteins are a kind of RBP that features cytoplasmic and nucleolar distribution and, like most other RBPs, they have broad and diverse roles in RNA regulation. The expression of CELF1 and CELF2 is detected in almost all tissues; however; their patterns can change in different developmental and differentiation stages. CELF3-6 are mostly present in neurons, and CELF6 can also be found in the kidneys and the testes [5]. All six members of this family feature three RRMs (RNA Recognition Motifs) with unique divergent linker domains, which let them attach to downstream targets [6]. CELF proteins are involved in RNA splicing through binding to the upstream or downstream intron of an alternative exon to mediate exon skipping or inclusion. They are also abundant in the 3′UTR of transcripts associating with mRNA decay or translation [7,8]. 

Non-coding RNAs of various kinds can directly or indirectly target CELF proteins; additionally, these RBPs can significantly alter non-coding RNAs in many ways, forming the noncoding RNA/CELFs regulatory axis. Long noncoding RNAs (lncRNAs), micro RNAs (miRNAs), and circular RNAs (circRNAs) are three forms of noncoding RNAs whose expression may be modified, and that can either promote or prevent carcinogenesis. LncRNAs with a length of longer than 200 nucleotides are transcribed by RNA polymerase II and preferentially stay in the nucleus to perform the regulation of gene expression and splicing [9]. MiRNAs are typically 20–23 nucleotides long, with the seed region for base pairing to target mRNAs located in seven or eight nucleotides at the 5′ end of the miRNA. The expression of a target mRNA is expected to bear a modest or strong reduction upon the binding of one miRNA or multiple miRNAs, respectively [10]. Lastly, circRNAs can be found broadly in almost all cells and tissues; they are mainly distributed in the cytoplasm and, rarely, in the nucleus. They can function as RNA sponges, nucleators of multiprotein complexes, or even transcriptional regulators [9]. 

The noncoding RNA/RBPs’ regulatory axis has its own complexity, given the many roles and interactions of both RBPs and noncoding RNAs. Each side of this axis has regulatory implications for the other and many additional objectives. The various activities of CELF RBPs, their regulation, and their significance in the noncoding RNA/RBPs’ regulatory axis are highlighted in this review.

## 2. General Characteristics of the CELF Protein Family

So far, six CELF RBPs have been discovered, with a nomenclature ranging from CELF1 to CELF6. Based on their phylogenetic structure and expression pattern, these members may be divided into two subfamilies: CELF1-2 and CELF3-6. CELF1 and CELF2 are extensively expressed in numerous tissues throughout the human body, but CELF3 through 5 are mostly found in the neurological system. CELF6 is most often found in the nervous system, the kidneys, and the testes [5,11,12]. All six members have a similar structure that includes three RRMs (RNA Recognition Motifs), two in the N-terminal and one in the C-terminal, as well as a divergent linker region that can improve RBP binding affinity through conformational variations. The nuclear localization signal, which comprises a lysine/arginine-rich region in the C terminal and a region important for nuclear export in the linker domain, are two other key structures [6,13,14]. A serine/threonine-rich phosphorylation area within the CELF1 and CELF2 linker domains could change their affinity for binding to target mRNAs [15,16]. However, there are no reports regarding the phosphorylation of CELF2 at this site. Apart from phosphorylation, the acetylation of the lysine436 residue inside the third RRM motif has been shown to influence CELF protein–protein interactions and mRNA affinity [17,18]. There are few studies in relation to the evolution of the CELF family members; however, their presence in xenopus, zebrafish, and mammals is clear evidence of their ontological importance and their conservation. During embryonic development CELF1 localizes in the vegetal pole of zebrafish and CELF2 in the animal pole [19,20]. This compartmentalization suggests that each protein performs different functions and, therefore, that they must have separated very early in evolutionary terms. Future works about the evolution and function of CELF proteins would clarify how and why these proteins arose.

By generating different isoforms, the alternative splicing process modifies the structure, expression, cellular location, and functional properties of CELF family members. For example, CELF2 features three promoters that create three distinct isoforms, each of which features a unique N-terminal sequence and 5′UTR [21]. CELF2 loses its RNA binding capacity when exon 14 is skipped; moreover, exon 9 skipping is reported for CELF4 [21,22]. Alternative splicing on 3′UTRs and C termini of CELF2 and CELF5 may affect their regulation and function [6]. Figure 1 illustrates the structure and conservation of all six proteins of the CELF family. Reported phosphorylation and acetylation sites are obtained from PhosphoSitePlus [23].

CELF proteins are found in the cytoplasm and nucleus, but they can also be found in stress granules, RNA neuronal granules, and on the PNC (peri-nuclear compartment) under specific circumstances [6,24,25]. As observed most clearly in skeletal muscle loss [26], CELF action remarkably depends on cellular location. CELF2 appears to be primarily or entirely nuclear, unlike CELF1, which is expressed in both the nucleus and the cytoplasm. As a result, despite the fact that they may bind to similar sequences, CELF2 and CELF1 are predicted to have distinct impacts on RNA processing [7,27].

CELF RBPs are primarily involved in alternative splicing and transcript editing in the nucleus. Members of this family are all potential global alternative splicing regulators by generally recognizing CUG and UG-repeat elements [28,29]. According to a genome-wide interaction analysis between CELF1 and its target RNAs, this RBP’s distribution at exon-intron borders, 3UTR, and even exonic regions can comprehensively affect alternative splicing events [8,29]. However, more complex CELF1 binding sites are occasionally required for more specialized splicing processes. An alternative 3′UTR splicing site with two CELF1 binding sites separated by a long intron is required to regulate the mPGES-1 isoforms [30]. Similarly, CELF2 has been identified as an alternative splicing factor for a high number of genes [31]. As observed for exon 6 of LEF1 (Lymphoid Enhancer Binding Factor) and exon 2 of MKK7 (Mitogen-Activated Protein kinase 7) transcripts, CELF2 binding to a downstream or upstream region of an alternate exon can result in that exon being either included or excluded, respectively [7]. CELF2 is additionally a regulator of 3′ UTR intron retention by competing with the U2-auxiliary factor (U2AF65) subunit. Notably, CELF2 mediates the 3′ UTR intron retention of its own transcript, as well as many others [32]. Lastly, CELF2 has been found to be involved in another nuclear function: blocking the C to U editing of apolipoprotein B (APOB) RNA, a process that may be stimulated by CELF2 knockdown [33]. 

In the cytoplasm, CELF proteins are involved in mRNA stability, translation, alternative polyadenylation (APA), and pre-miRNA maturation [12,32,34,35,36]. CELFs regulate mRNA stability across the genome, although only specific, well-studied events are discussed in this section. CELF1 controls mRNA stability by identifying “UGUUUGUUUGU” element (GU-rich element, GRE) or GU-repeat sequences in the 3′UTR, which mechanistically appear to rely on CELF1 interaction with ribonucleases, such as poly(A)-specific ribonuclease (PARN) [8,26]. PARN 3′-exoribonuclease destabilizes transcripts by removing poly(A) tails at the 3′ end [37]. On the other hand, CELF2 stabilizes the cytoxygenase-2 (COX2) transcript through a different method, which involves binding to A/U-rich regions in the 3′UTR of mRNA [5]. CELF6 has been shown to bind predominantly to certain 3′UTR regions, resulting in a substantial decrease in mRNA levels [12]. 

CELF proteins have been found to regulate widespread translational activation or inhibition by binding to target transcripts’ 5′UTR or 3′UTR [38]. For instance, the timed translation of Elavl4 (Embryonic Lethal, Abnormal Vision, Drosophila-Like 4) isoforms depends on distinct 5′UTRs controlled by CELF1 during neuronal development [39]. Moreover, CELF1 is able to regulate P21 mRNA stability by binding to its 5′UTR upon proteasome inhibition [40]. Binding through 3′UTR, CELF2 can inhibit the translation of MCL1 (Apoptosis Regulator, BCL2 Family Member) anti-apoptotic factor [41]. Another example is the circularization of SHMT1 (Serine Hydroxymethyltransferase 1) mRNA and its internal ribosome entry site (IRES)- mediated translation, which is provided with CELF1-hnRNPH (Heterogeneous Nuclear Ribonucleoprotein H1) proteins, implying a 5′/3′ interaction in the SHMT1 transcript [42]. Interestingly, following CELF2 depletion, hnRNPC (Heterogeneous Nuclear Ribonucleoprotein C) translation efficiency is shown to be reduced. This is mostly due to the hnRNPC mRNA’s lower translation elongation efficiency [34]. Further pathway study indicates that the knockdown of the CELF1 decreases the expression of some ribosomal proteins in the 40S and 60S subunits, indicating a broader function of the CELF1 in translation [29]. 

CELF proteins are known to regulate alternative polyadenylation (APA) via two mechanisms. Firstly, CELF1 has been shown to bind to c-FOS (Fos Proto-Oncogene, AP-1 Transcription Factor Subunit) and TNFα (Tumor Necrosis Factor-Alpha) transcripts in a HeLa cell extract and promote deadenylation by interacting with PARN, indicating that CELF1 helps both deadenylation and mRNA stability [35]. Secondly, CELF2 prevents polyadenylation machinery from binding to RNA by competing with CFI (cleavage factor I) and CstF (cleavage stimulatory factor) components in a cell signal-dependent way [32]. 

Recently, CELF2 was reported as a potential regulator of pre-miR155 maturation. CELF2 is reported to be associated with IL10 (Interleukin-10) to inhibit the maturation of pre-miR155 by blocking KSRP (KH-Type Splicing Regulatory Protein). In this way, it can be considered as negative regulator of miR155 [36]. Similar results are reported for miR140, where CELF1/2 binding interferes with DICER binding to miR140 and thus inhibits its maturation [43]. 

Finally, CELF RBPs are present in stress granules, neuronal RNA granules, and the peri-nucleolar compartment (PNC). Stress granules (SGs) are dynamic tiny ribonucleoprotein condensates playing functional roles in mRNA processing and degradation and are found in all eukaryotes during cell stress conditions [44]. CELF1 and CELF2 have been found to be required for the trafficking, stability, and translation of p21/CDKN1A and COX-2 to SGs respectively [40,45]. Additionally, CELF4 associates with very large neuronal RNA granules to mediate neurotransmission by regulating the stability, translation, and/or localization of many downstream target mRNAs [25]. 

PNCs are specialized subnuclear organelles located adjacent to the nucleolus. They are associated with various solid malignant tumors. This region contains transcripts produced by RNA polymerase III, small nuclear RNAs, and many RBPs, including CELF1 [46,47]. In PNCs, CELF1 associates with the RNA component of RNAse MRP (mitochondrial RNA processing) and PTBP (polypyrimidine tract binding protein), interacting with newly synthesized RNA. However, CELF1 enrichment in PNCs may restrict it from performing its cytoplasmic functions [6,48]. 

In consequence, CELF members are highly divergent, pleiotropic RBPs that are responsible for global alterations in numerous cell types and disease situations [5]. However, CELF proteins are not the only multifunctional RBPs. In two previously published studies, various RBPs influencing different cancers were described [49,50]. The principal functions of CELF family proteins are depicted in Figure 2. Furthermore, they are controlled by numerous biological elements that are discussed specifically in relation to cancer in this article. 

## 3. CELF Targets in Cancer

The wide-ranging targets and diverse functions of CELF proteins in cancer have elevated them as prominent players with critical roles in divergent biological pathways. As a result, they might be regarded as prospective prognostic and therapeutic targets. However, present studies on the CELF family, particularly members 3–6, are quite sparse, and it appears that additional studies in this area are required in the future. Notably, despite the structural resemblance between CELF family members, particularly between CELF1 and CELF2 (>90% similarity in their RNA-binding domains), their functions are quite different, notably in cancer [6].

### 3.1. CELF1

CELF1, also referred to as CUG-binding protein 1 (CUGBP1), is found mostly in the heart, skeletal muscle, and brain, and has been linked to myotonic dystrophy type 1 [51,52,53], muscle wasting [26], and Alzheimer’s disease [54,55], as well as multiple cancers. CELF1 preferentially binds to 3′UTRs and mediates a wide range of mRNA degradation and alternative splicing events. Interestingly, the binding of CELF1 to the upstream intron of an alternative exon promotes exon skipping, whereas binding to the downstream intron enhances exon inclusion. The same mechanism has already been discovered in the proteins NOVA (NOVA Alternative Splicing Regulator), FOX2 (RNA Binding Fox-1 Homolog 2), and PTB (Polypyrimidine Tract Binding Protein 1) [56]. 

Genome-wide interactions of CELF1 proteins were previously detected in HeLa cells utilizing RIP-seq (RNA immunoprecipitation sequencing) and CLIP-seq (cross-linking and immunoprecipitation sequencing). In HeLa cells, global alternative splicing events with a substantial increase in exon inclusion under the regulation of CELF1 have been identified by mapping the RIP-seq reads [8]. By combining CLIP and high throughput sequencing, a large number of CELF1 target clusters were reported, most notably LMO4 (LIM Domain Only 4), BAG1 (BAG Cochaperone 1), PKM (Pyruvate Kinase M1/2), SIX5 (SIX Homeobox 5), and DMPK (DM1 Protein Kinase) transcripts in human HeLa cells [57]. Previously, CELF1-bound melanoma-enriched transcripts, including the oncogene DEK, were discovered using systems analysis, and CELF1 has been described as a risk factor for overall patient survival [58]. More than 1000 mRNAs associated with cell proliferation, angiogenesis, and signal transduction are differently regulated by CELF1 in oral squamous carcinoma cell lines. The depletion of CELF1 reduces proliferation and increases apoptosis in oral squamous cell carcinoma where CELF1 associates directly with the 3′UTR of BAD (BCL2 Associated Agonist Of Cell Death), BAX (BCL2 Associated X, Apoptosis Regulator) and JunD mRNAs and mediates their rapid decay [59,60]. The results are the same in non-small-cell lung cancer, where the downregulation of CELF1 induces an increase in the protein levels of cyclin D1, BAD, BAX, Jun D, and E-cadherin, indicating a function for CELF1 in the EMT (epithelial to mesenchymal transition). EMT is one of the critical stages involved in cell progression, invasion, and metastasis [61]. CELF1 is a key component of the EMT’s post-transcriptional regulatory networks. Thus, CELF1 overexpression is linked to disease stage, differentiation, and a poor prognosis in individuals with non-small-cell lung cancer [38,62,63,64]. 

Importantly, phosphorylation makes substantial changes to CELF1 functions. Akt kinase (also known as protein kinase B or PKB) phosphorylates CELF1 at Ser28, whereas Cyclin D/cdk4/6 phosphorylates Ser302 residue. This post-translational alteration can improve the protein’s affinity with specific mRNA substrates and influence CELF1’s capacity to interact with translation factors [15,65]. Notably, both Akt and CDK4/6 protein kinases are considered important targets for cancer treatment [66,67]. CELF1 target transcripts change substantially across malignant T-cell lines and primary human T cells. This is mostly due to CELF1 phosphorylation in malignant T cells, which inhibits it from binding to mRNAs, resulting in the rapid degradation of different cell proliferation suppressors [16,68]. In hepatoblastoma, phosphorylation converts CELF1 in its oncogenic form. It has been suggested that CELF1 is a tumor suppressor protein when it is dephosphorylated at Ser302, but that it is an oncogene when it is phosphorylated at this residue; the upregulation of its phosphorylated form has been reported in mouse models [69,70]. 

In glioma, the increased expression of CELF1 is positively correlated with glioma grading and inversely correlated with patient survival, making it a novel prognostic predictor for glioma patients. SiRNA (small-interfering RNA)-mediated knockdown of CELF1 inhibits the glioma cell cycle progression and proliferation. In glioma cells, the tumor suppressor CDKN1B (Cyclin Dependent Kinase Inhibitor 1B) has been identified as a CELF1 target. CELF1 interaction with CDKN1B protein, but not with mRNA, causes this cyclin-dependent kinase inhibitor to be suppressed, but the mechanism of this inhibition is unknown [71]. 

In breast cancer, CELF1 promotes exon 11 exclusion of the INSR (insulin receptor gene), promoting the production of the pro-tumorigenic IR-A splicing isoform [72]. CELF1 that is induced by TGF-β (Transforming Growth Factor Beta) can directly bind and trigger a set of translationally controlled EMT drivers, promoting breast cancer tumor growth [38]. 

CELF1 and ETS2 are both upregulated in colorectal cancer cells and can promote tumorigenesis. Luciferase reporter and ribonucleoprotein immunoprecipitation assays show that CELF1 contributes to ETS2 overexpression by binding to its 3′UTR. ETS2 overexpression has been found in a number of human malignancies [73]. 

Lastly, the CELF1-mediated stability of p21 mRNA in stress granules and after bortezomib chemotherapeutic drug therapy results in apoptotic resistance [40]. Moreover, lowering CELF1 is the main mechanism of tumor suppression of breast cancer cells with a combination of glycyrrhetinic acid and doxorubicin [74]. 

### 3.2. CELF2

CELF2, a pleiotropic member that regulates many cancer-related genes, is primarily downregulated in tumors [75]. A pan-cancer analysis in 2021 demonstrated strong evidence of a correlation between CELF2 upregulation with better prognosis in multiple tumors, particularly in breast and lung cancers, where it was markedly associated with many immune checkpoint molecules. In this study, CELF2 was presented as a more promising biomarker than five prevalent biomarkers, including PD-1, PD-L1, CTLA-4, CD8, for selecting immunotherapy-sensitive patients [76]. 

In breast cancer, CELF2 overexpression suppresses proliferation and invasion, and inhibits tumor growth and angiogenesis. It probably achieves this by decreasing the expression of N-cadherin (N-cad), CD34, and NFATc1 (the nuclear factor of activated T cells, also known as NFAT2). The NFAT family of proteins are a group of transcription factors found in the cytoplasm of T lymphocytes and their activation stimulates the progression of hematological malignancies, as well as that of solid tumors [77]. Previously, it was revealed that CELF2 suppression by promoter hypermethylation in breast cancer cells induces modulations of intron retention patterns and isoform imbalance in many target genes. These modifications are mainly evident in genes involved in cancer critical pathways, such as autophagy, apoptosis, EGFR (Epidermal Growth Factor Receptor) signaling, and metastasis [78]. These findings may imply the critical role of CELF2 as a tumor suppressor in breast cancer. 

Further evidence can be found in non-small-cell lung cancer, where CELF2 is capable of inhibiting tumor development by antagonizing PREX2 (Phosphatidylinositol-3,4,5-Trisphosphate Dependent Rac Exchange Factor 2)’s oncogenic impact and regulates PI3-K signaling [79]. The overexpression of CELF2 is associated with better overall survival-related alternative splicing events in lung adenocarcinoma [80]. CELF2 increases the stability of the tumor suppressor FAM198B in ovarian cancer by binding to AU/U-rich elements in the 3′UTR regions. CELF2 can also influence ovarian cancer cell proliferation, migration, and invasion, making it more susceptible to cisplatin [81]. Interestingly, the mRNA stability and trafficking of the COX-2 (Cyclooxygenase-2) tumor promoting agent to cytoplasmic SGs depends significantly on CELF2 RBP [45]. 

CELF2 expression is a potential prognostic biomarker of multiple cancers. Three splicing factors, CELF2, ESRP2 (epithelial splicing regulatory protein 2), and SRSF5 (serine- and arginine-rich splicing factor 5), which are substantially downregulated in hepatocellular carcinoma samples, were found to be responsible for splicing dysregulation in hepatocellular carcinoma [82]. In this carcinoma, STYXL1 (Serine/Threonine/Tyrosine Interacting Like 1) promotes malignant progression via the downregulation of CELF2 and the activation of the PI3K/Akt pathway [83]. In addition, CELF2 has been identified as one of the hub splicing factors of prognostic splicing events, suggesting that these splicing factors might be used to treat glioma [84]. Moreover, in oral squamous cell carcinoma, CELF2 is identified as a promising prognostic biomarker [85].

Conversely, in gastric cancer, the expression of CELF2, along with BAG2 (BAG Cochaperone 2), RBFOX2 (RNA Binding Fox-1 Homolog 20), and PTBP2 (Polypyrimidine Tract Binding Protein 2) splicing factors are associated with poor prognosis and alternative splicing events [86]. 

CELF2 plays a critical role during T cell activation. CELF2 and hnRNPC (Heterogeneous Nuclear Ribonucleoprotein C) are trans-acting factors in T cells that can positively regulate each other. This regulation occurs when HNRNPC promotes CELF2 mRNA transcription and CELF2 regulates hnRNPC mRNA translation. This has a significant impact on downstream targets, particularly overlapping sets of splicing events, especially in the TRAF3 (TNF Receptor Associated Factor 3), LEF1 (Lymphoid Enhancer Binding Factor 1), and MKK7 genes [34,87]. Many splicing processes during T-cell stimulation and human thymic T-cell development have previously been shown to be correlated to CELF2 expression [88]. The near-collaboration of CELF2 with the JNK signaling pathway controls splicing during T-cell activation [89]. Notably, the presence of T cells in cancer lesions correlates with a better patient prognosis in a variety of human malignancies [90], highlighting CELF2 as a potential tumor progression inhibitor.

Ultimately, CELF2 may be beneficial in cancer therapy. Curcumin suppresses pancreatic tumor development by raising the expression of CELF2, which prevents COX-2 and VEGFA (Vascular Endothelial Growth Factor A) mRNAs from being translated [91]. Similarly, the upregulation of CELF2 increases the response of pancreatic cancer cells to chemotherapy [92]. The activation of apoptosis and autophagy by ionizing radiation (IR) causes cancer cells to die. The upregulation of CELF2 enhances IR-induced autophagy in colorectal cancer, which has been revealed to be a significant role in this process [93]. 

### 3.3. Other CELF Members

Members 3-6 of the CELF family are also key players in global RNA processing modification; however, little is known about their impact in cancer [5]. Two studies have shown that CELF3 may play a role in cancer. The upregulation of CELF3 is reported to be induced by HPV16 E6 and is thought to be crucial for downstream RNA splicing changes, such as exon skipping [94]. Furthermore, in patients with colorectal cancer metastasis, different patterns of expression and copy number variations are observed for CELF3. As a result, this RBP, along with three others, APOBEC3G (Apolipoprotein B MRNA Editing Enzyme Catalytic Subunit 3G), EEF1A2 (Eukaryotic Translation Elongation Factor 1 Alpha 2), and EIF5AL1 (Eukaryotic Translation Initiation Factor 5A Like 1), are reported as potential metastasis-associated factors in colorectal cancer [95]. 

CELF4 is a broadly expressed gene. However, reports on its expression are conflicting [5]. The first evidence of an association between CELF4 and cancer comes from two studies on the genetic variant rs1786814, according to which the SNP contributes to chemotherapy-related cardiac dysfunction as a significant side effect of anticancer therapy. This variant can probably cause the loss of a donor splice site in CELF4 [96,97]. CELF4 as later shown to be hypermethylated in endometrial cancer, making it a viable target for cancer screening in cervical smears [98]. Recently, an analysis of sequencing data from the two related hypothesized colorectal cancer predisposition haplotype carriers on the 18q12.2 locus revealed a rare intronic variant of the CELF4 gene, rs568643870, that is significantly associated with colorectal cancer and segregates with this malignancy in other members of the linked pedigree [99]. 

Apart from this study, colorectal cancer is the most studied malignancy in relation to the CELF4 expression level and mutations. In this context, CELF4 has been identified as one of the reproducible prognostic RBPs in multiple investigations; however, the results are controversial [100,101,102,103]. CELF4 downregulation in colorectal cancer samples is detected through analysis of the Cancer Genome Atlas database (TCGA). This gene, along with other RBP genes, has created a survival-related network with remarkable prognostic value [100,101,102]. On the other hand, the predictive study of the characteristics of RBPs in colorectal cancer highlights the significant role of CELF4 as an overexpressed transcript in highly metastatic SW620 cell lines [103]. 

CELF5 is another member of the developmentally expressed CELF family that is present primarily in the fore, middle, and hind brain and performs specific functions in mRNA splicing and translation [104,105]. CELF5 was previously described as a modifying factor of motor-neuron disease-relevant pathways [106]. In 2018, researchers revealed that CELF5 overexpression is connected to enhanced HCMV (Human Cytomegalovirus) DNA synthesis via a direct interaction with viral UL141, a protein that increases the intracellular retention of death receptors to shield virus-infected cells from NK cell-mediated killing [107]. HCMV is a prevalent virus with onco-modulatory properties that can be associated with malignant tumors [108]. CELF5 is the most significant splicing factor associated with overall survival splicing events, regulating germ cell-specific gene 1-like (GSG1L) isoform diversity, according to a recent study of RNA sequence and alternative splicing events from glioblastoma multiforme samples. Nonetheless, it is unclear whether CELF5 has a positive or negative regulatory connection with carcinogenesis, although this network may have a major impact on prognosis [109]. 

CELF6, first detected in 2004 as a new member of the CELF family, has a widespread expression in the brain from fetuses to adults. Furthermore, its expression is identified in the nervous system, kidneys, testes, and, to a lesser extent, in most other tissues. Like other members of the CELF family, CELF6 shuttles between the cytoplasm and the nucleus, but is predominantly located in the cytoplasm [104,110]. CELF6 is involved in post-transcriptional control rather than alternative splicing, according to CLIP-seq and large parallel functional analyses in the brain. Changes in CELF6 levels have been linked to protein abundance and localization, particularly in the FOS (Fos Proto-Oncogene, AP-1 Transcription Factor Subunit) and FGF13 (Fibroblast Growth Factor 13) targets examined [12]. CELF6 is capable of regulating U2 Auxiliary Factor 2 (U2AF2) by improving its binding and thus modulating the splicing process in the U2AF2 target genes [111]. The binding of U2AF2 to a uridine/cytidine-rich sequence element upstream of the 3′ splice site is an important regulatory step and is essential to recruit the small nuclear ribonucleoprotein (snRNP) U2, a subunit of the spliceosome [112,113]. Mutations in U2AF2 have direct effects on neoplastic transformation, and its stabilization and overexpression are significantly associated with worse prognosis for cancer patients [114,115,116].

The first observations on the association of CELF6 with cancer came in a study on susceptibility to cervical cancer, in which the minor allele “C” of rs4777498 in the CELF6 gene accounted for an increased risk of this malignancy [117]. Interestingly, by binding to the 3′UTR of the p21 and FBP1 (Fructose Bisphosphatase 1) transcripts, CELF6 is able to stabilize their mRNAs. In colorectal cancer, depending on the p53 and/or p21, CELF6 can induce the arrest of the G1 phase. The multifunctional p21 protein plays a significant part in cell cycle progression, cell senescence, apoptosis, and transcriptional regulation [118,119]. Moreover, the stabilization of FBP1 mRNA by CELF6 leads to suppressed cell proliferation, cell migration, and cell invasion in triple-negative breast cancer [120]. FBP1 performs a tumor suppressor function by limiting glucose uptake and glycolysis [121].

Lastly, exposure to cadmium through smoking is implicated in the DNA methylation modifications of oncogenes, tumor suppressor genes, and genes related to inflammation, making it one of the important risk factors for cancer [122,123]. A recent study in 2020 showed that elevated cadmium concentrations in the blood are involved in the overmethylation of the cg11314779 site within the CELF6 gene, which is probably associated with its silencing [124,125]. CELF6 appears to play an essential role in tumor suppression via global post-transcriptional changes.

## 4. CELF Regulation by Noncoding RNAs

Three types of noncoding RNA, including miRNAs, lncRNA, and circRNAs with protein-coding genes, can construct complicated networks regulating the expression of many downstream pathways [9]. Many noncoding RNAs that target CELF members have recently been discovered, and they appear to have a significant influence on the development of cancer. The regulatory networks for CELF1 and CELF2 are illustrated in Figure 3A,B.

MicroRNAs are 21- to 24-nucleotide RNAs, tiny molecules that influence the stability and translation of over 18,000 potential mRNA targets favorably or negatively [126,127]. Mir-330-3p has been shown in certain studies to be an inhibitor of carcinogenesis [128]. This tumor suppressor is reported to have low expression in glioma tissues and cells, and it is able to prevent proliferation and migration by directly binding to CELF1 and suppressing its expression [129]. In human lung tumors, miR-574-5p upregulation promotes tumor development in vivo and increases mPGES-1 expression by preventing CUGBP1 binding to the mPGES-1 3′UTR (decoy mechanism), which leads to an enhanced alternative splicing mPGES-1 and the generation of a novel 3′UTR isoform [130]. The specificity of miR-574-5p/CELF1 regulation on mPGES-1 expression in human lung cancer cells is discovered in a recent proteomic analysis. Patients with lung cancer who have high miR-574-5p levels may benefit significantly from the pharmacological inhibition of PGE2 (Prostaglandin E Synthase 2) formation [30]. It is likely that miRNA-mediated CELF1 inhibition has tumor suppressor effects; however, additional research is needed for different malignancies.

According to numerous studies, CELF2, a tumor suppressor, is also downregulated by a number of miRNAs. In gastric cancer, the inhibition of CELF2 by miR-615-3p enhances proliferation and migration while blocking apoptosis [131]. However, depending on its expression levels, mir-615-3p can have tumor suppressive or carcinogenic effects in some other malignancies. [132]. In glioma cells, CELF2 is a potential target of miR-20a and miR-95-3p, in which the overexpression of these two miRNAs is positively correlated with cell proliferation and invasion [133,134]. In spinal cord glioma tissues, the overexpression of miR-106a-5p exerts its oncogenic function by decreasing CELF2 expression [135]. Exosomes are extracellular vesicles that transport information between healthy cells as well as in tumor microenvironments, allowing cell-to-cell communication. Exosomes from hypoxia colorectal cancer cells transport miR-210-3p to normoxic tumor cells, promoting G1-S cycle transition and proliferation while inhibiting apoptosis by downregulating CELF2 expression [136]. Brucein-D, a drug that targets miR-95, reduces hepatocellular carcinoma (HCC) cell proliferation in vitro and tumor growth in vivo. CELF2 has also been discovered to be a miR-95 downstream target [137]. 

The miRNA miR-375, which can regulate CELF6, is thought to be associated with pancreatic cancer and breast cancer [138,139,140,141]. In breast cancer, higher levels of miR-375 were expressed in ER-α-positive, where it was a critical factor in cell proliferation and an early event in tumorigenesis [142]. A single nucleotide variation, rs4777498, in CELF6, regulated by miR-375, was previously found to be associated with cervical cancer susceptibility [117]. The upregulation of miR-375 is a key driver of cell proliferation and patients with miR-375 overexpression have a higher probability of local relapse in early breast cancer [140].

LncRNAs are a class of RNA transcripts with more than 200 nucleotides, with secondary and three-dimensional structures that enable them to regulate their targets as cis or trans-acting elements. They are mainly present in the nucleus; however, they can also be found in cytoplasm. By regulating epigenetic, transcriptional, and post-transcriptional modifications, lncRNAs play a key role in a variety of cancers [143]. In hepatocellular carcinoma, the LncRNA BACE1-AS, which plays a crucial role in carcinogenesis, is overexpressed. BACE1-AS inhibits miR-377-3p, and CELF1 is a downstream target of miR-377-3p that serves as an oncogene [144]. In nasopharyngeal carcinoma, LncRNA AFAP1-AS1 can enhance cancer growth by affecting the miR-497-5p/CELF1 axis [145].

The RNA network also has CELF2 as a target. In acute lymphoblastic leukemia (ALL) and acute myeloid leukemia (AML) patients and cell lines, the lncRNA small nucleolar RNA host gene 16 (SNHG16) overexpresses, and by lowering CELF2 mRNA stability, this lncRNA enhances proliferation and migration [146]. In hepatocellular carcinoma, the lncRNA CRNDE (differentially expressed colorectal neoplasia) is considerably overexpressed and is associated with poor clinical outcomes. Mechanically, CRNDE is implied in the inhibition of tumor suppressor genes, including CELF2 and LATS2 (large tumor suppressor 2) [147]. Overexpression of the lncRNA RHPN1-AS1 is associated with enhanced cell viability, proliferation, migration, and invasion in nasopharyngeal carcinoma by targeting CELF2, resulting in an increased mTORC1 signaling pathway [148]. Finally, the LncRNA CCDC26 binds to CELF2, upregulating the expression of circRNA ANKIB1. By sponging miR-195-5p, the overexpression of circRNA ANKIB1 increases PRR11 (Proline Rich 11) protein production, activating the PI3K/AKT and NF-B pathways. The overexpression of the LncRNA CCDC26 in myeloid leukemia enhances cancer development in this way [149].

In summary, CELF proteins play a key role in the noncoding RNA/RBP regulatory axis as they transform noncoding RNA instructions into a variety of downstream signals.

## 5. Noncoding RNA Regulation by CELF

The flip side of the noncoding RNA/RBP coin is RBPs’ ability to influence noncoding RNA expression and functionality. This is summarized in Figure 3C. The 3′UTR elements play a regulatory role as the primary site of miRNA binding and function, as well as influencing mRNA destiny in terms of translation, stability, and subcellular localization [32]. Meanwhile, the global distribution of CELF members can be found in 3′UTR regions. In addition, intergenic regions that are involved in the process of non-coding RNA biogenesis are also enriched for CELF proteins [7,8,12]. RBPs have recently been found to improve miRNA targeting (MT) by opening mRNA secondary structures [2]. Notably, the “UUUGUUU” motifs, which are similar to CELF binding sites, are enriched adjacent to miRNA binding sites, and their presence tends to potentiate miRNA activity [150]. 

Cooperative or antagonistic RBP–RBP interactions should not be overlooked in this axis, especially in the case of miRNA regulation. CELF2 appears to play an important function in the pre-miRNA maturation process as it can potentially repress RBFOX2 (RNA Binding Fox-1 Homolog 2) mRNA and protein levels in human cells. These two RBPs have an evolutionary conserved antagonistic relationship that regulates splicing processes in a variety of signaling pathways and transcription factors [31]. The Rbfox proteins’ binding site is assumed to be the conserved GCAUG sequence, which allows them to attach to the terminal loops of miR-20b and miR-107 precursors and inhibit their nuclear processing. MiR-107, in turn, targets Dicer mRNA’s 3′UTR and downregulates it in a way that correlates to the invasiveness of breast cancer cell lines [151]. Rbfox RBPs can influence the expression of downstream targets in cancer by regulating mature miRNA levels. Additionally, RBFOX2 was shown to be one of the specific miRNA hairpin binding proteins binding at multiple targets [43,152]. 

CELF2 was recently discovered to be a possible regulator of pre-miR155 maturation, with CELF2 knockdown reducing IL10’s ability to block pre-miR155 processing. CELF2 binding to pre-miR155 may hamper DICER’s capacity to bind to and cleave pre-miR155 [36]. Other proteins required for pre-miR155 processing, such as KH-type splicing regulatory protein (KSRP), may also be blocked by CELF2 binding to pre-miR155 [153]. Similar findings have been reported for pre-miR140, in which CELF1/2 binding inhibits pre-miR140 processing by interfering with DICER binding [43]. Another RBP, HuR (also known as ELAV Like RNA Binding Protein 1), is thought to enhance miRNA targeting by attracting AGO (Argonaute) or opening the secondary structure to make AGO more accessible. This RBP can also prevent miRNA targeting by competing against AGO [154]. Competition between CELF1 and HuR for binding to 3′UTR is possible, although this association has only been observed in Myc mRNA translation [155].

The nuclear aggregates of CELF1 and the tumorigenic lincRNAs (long intergenic non-coding RNAs) NEAT1 and NEAT2/MALAT1 may also help to explain how CELF members interact with noncoding RNAs. These aggregates might indicate a function for CELF1 in lincRNA biology, although lincRNAs could also operate as “CELF1 sponges” [57,156]. CELF1/lncRNA combinations with strong affinity have also been seen in colorectal cancer, where CELF1-TNBL aggregates occur in non-perinucleolar areas upon demethylation events [157]. 

As a result, evaluating all the methods through which CELF members can widely control noncoding RNAs makes it impossible to determine who has the upper hand in this game.

## 6. Pharmacological Insights of CELF/Noncoding RNA Axis

In a novel cancer therapy approach, small molecules have shown pharmacological potential as anticancer drugs to inhibit particular targets in cancer cells [158]. Dephosphorylated CELF1 was shown to be degraded by Gank (Gankyrin oncogene). The Cjoc42, a tiny Gank-inhibiting compound, stimulates CELF1 expression, improves chemo-sensitivity, and reduces liver cancer cell proliferation in vitro [70,159]. Another natural product, Fraxinellone, notably inhibits the mRNA expression of CELF1 in models of kidney and liver fibrosis [160]. Remarkably, cancer-associated fibrosis is linked to modifications in the tumor microenvironment impacting cancer behavior [161]. Another compound that downregulates the expression of CELF1 is the piperazine derivative BK10007S, which induces apoptosis through this inhibition in hepatocellular carcinoma cells [162]. To our knowledge, there are no reports regarding other CELF family members. In the same context, the involvement of noncoding RNA biology in the cancer background has become a source of new diagnostic and therapeutic tools and has been widely studied. Some examples are listed in Table 1.

It is notable that some miRNAs regulating CELF have the same profile of expression in different cancer types in response to different drugs, as in the case of miR-106, others vary according to the tested drug in the same cancer type (miR-210), and, in a third scenario, there are miRNAs that are differentially expressed in the treated group and not in the control group. CELF2 has been shown to be a miR-95 downstream target and Brucein-D, a drug that targets miR-95 and reduces hepatocellular carcinoma (HCC) cell proliferation in vitro and tumor growth in vivo [137]. Drugs that target miRNA might be used with other chemotherapeutic drugs in the future, potentially altering drug resistance concerns in cancer therapy.

Apart from BACE1 inhibitors, which have been widely studied due to their role in pathologies such as Alzheimer disease and some malignancies [178], there is a lack of information regarding pharmacological inhibitors of lncRNA. Because a substantial body of data suggests that lncRNAs play a role in the pathogenesis of several cancer types, lncRNAs can be used as prognostic biomarkers and therapeutic agents for malignancies. For example, SNHG16 binds to endogenous miRNAs, resulting in the aberrant expression of downstream genes or affecting signaling transductions. This lncRNA is upregulated in various tumor types and it is associated with indicators of poor prognosis [179]. Finally, RNA-PROTACs are a novel type of chimeric oligonucleotide that directly dock into an RBP’s RNA binding site and guide the RBP towards proteasomal destruction through a conjugated E3-recruiting peptide [180]. This newly proven drug therapy in cancer cells can be considered as promising for the pharmacological targeting of CELF proteins.

## 7. Concluding Remarks and Future Perspectives

The misregulation of CELF members and their targets in cancer is a good illustration of how RBPs may regulate a variety of pathways while performing various tasks. Noncoding RNAs can also be thought of as direct regulators of numerous targets, including CELF proteins. The determination of the significance of the CELF/noncoding RNA axis, particularly in cancer, has many obscure aspects due to the scarcity of studies on the role of CELF proteins in regulating noncoding RNA metabolism and function.

CELF proteins are considered to play oncogenic and gene suppressor roles in cancer development. Putative binding sites for these proteins are distributed throughout the genome: the presence of CELF1 has been found to be abundant in the 3′UTR region, the intronic region, and the coding DNA sequences of HeLa cells. Surprisingly, 14.8% of CELF1 binding sites were identified in the intergenic region of the genome, which mainly contains non-coding RNAs, such as miRNA, highlighting the importance of studying this region in much greater detail. Moreover, the presence of CELF1 at exon-intron borders suggests that CELF1 plays an important role in RNA splicing, as has been observed in HeLa cells, whose direct binding to CELF1 can influence 451 splicing events in HeLa cells [8]. Through the regulation of splicing events, CELF proteins could regulate the activity of a plethora of genes and their biological functions. 

The subcellular localization of CELF proteins, as discussed above, is another essential consideration, since the kind of CELF proteins functions, as well as the target molecules, are affected by distribution in different cellular compartments. CELF1 aggregation in PNC, which is more common in cancer cells, can limit the protein’s cytoplasmic activities [6,24]. Furthermore, CELF1 expression in the nucleus has a substantial influence on alternative splicing events, whereas its expression in the cytoplasm has only a little impact. This shift in subcellular expression pattern could have a dramatic impact on the disease condition [26]. With this background, third-level drug targeting, which can directly deliver and target within specific intracellular sites, is recommended for specifically targeting CELF proteins accumulating in certain organelles. In this recent discovery, different organelle-specific targeting ligands, in conjugation with drugs or drug carriers, can be loaded in varied types of nanoparticles [181]. 

The collaborative or antagonistic interaction of CELF with other RBPs is another fascinating and complex feature of this family. CELF2 represses RBFOX2 mRNA and protein levels, resulting in antagonism between CELF and the RBFOX family. This relation might play an important regulatory role in the microRNA’s maturation process [31,43,152]. CELF2 and hnRNPC are trans-acting factors that can positively control each other’s translation, which has a significant impact on downstream target gene splicing [34,87]. Lastly, the circularization of SHMT1 mRNA and its internal ribosome entry site (IRES)- mediated translation is supplied by the collaborative interaction between CELF1 and hnRNPH in the SHMT1 transcript [42]. To discover all RBP–RBP interactions between two or more proteins across the genome, more in-depth RBP–RBP network analysis appears to be required.

Various miRNAs and lncRNAs have been shown to inhibit CELF members, including CELF1, CELF2, and CELF6, in a variety of malignancies. MicroRNAs and RBPs can work together to destabilize mRNA and stop it from being translated. They can also compete with one another, causing opposite effects on target mRNAs [182]. It is well known that microRNAs normally have more than one target in cells, but the induction of CELF proteins can reverse the situation, indicating the important role of CELFs in cancer progression [9,131,146]. Sometimes, more than one noncoding RNA is implicated in this axis, with an upstream lncRNA influencing a miRNA or circRNA, and widespread and drastic alterations occurring after targeting CELF proteins [144,149]. This might indicate that the number of actors on this axis will increase in the future.

In a study looking for RBPs that control microRNA levels by binding at identified pre-miR loci, researchers revealed that up to 92% of RBPs interact directly with at least one mir locus, with some interactions being cell-line-specific, which may indicate a role for RBP in miRNA metabolism [152]. According to a new discovery, the RNA structure becomes more open as the number of bound RBPs rises. When the RNA becomes less organized, miRNA-bound AGO may easily reach the MT sites by replacing the RBPs that were previously bound, resulting in productive MT [2]. However, to address the complex relation between RBPs and noncoding RNA production more conclusively, future efforts should be made to generate and analyze large-scale transcriptomic and proteomics datasets. 

CELF proteins are also prone to regulation by phosphorylation mechanisms. A decrease of CELF1’s binding capacity due to phosphorylation has been observed in cancerous T cells. CELF1 targets in malignant T cells involve cell cycle regulators, including those that regulate cell cycle progression throughout the G1, S, G2, and M phases, and components of cyclin-driven pathways that play important roles in cell development, differentiation, and tumorigenesis [68]. This sort of CELF1 alteration appears to be something that should be addressed in future target prediction research. The phosphorylation of CELF proteins adds more complexity to the CELFs’ regulatory network. This is a field that could provide a great opportunity to develop modulators of CELF protein activity, and should be addressed in the near future.

The cyclin-dependent kinase inhibitor p21CDKN1A induces cell cycle arrest following DNA damage by repressing the transcription of a group of cell cycle-regulatory genes; however, the expression of p21 is tightly controlled at the transcriptional and post-transcriptional levels [183]. In this context, several RBPs with crucial roles in p21 regulation have recently been extensively characterized [184]. However, CELF1 and CELF6 also play important roles in p21 regulation: CELF1 promotes p21 upregulation upon proteasome inhibition and CELF6 is required for p21 mRNA stabilization [40,118]. The different mechanisms of action of each CELF protein on the same target is also an important issue, not only for future drug design discoveries for cancer therapy, but also to determine the direct role of CELF family proteins in DNA damage. 

Consequently, this review attempted to gather information about the pleiotropic functions of members of CELF family of RBPs, with a focus on the noncoding RNA/CELF axis, in order to highlight the relevance of CELF balancing in cancer and, most importantly, to progress toward a better understanding of RNA processing alterations associated with cancer. This could provide a broad source of molecular tools that could be used to optimize the clinical care of cancer patients.

## Figures and Tables

**Figure 1 ijms-22-11056-f001:**
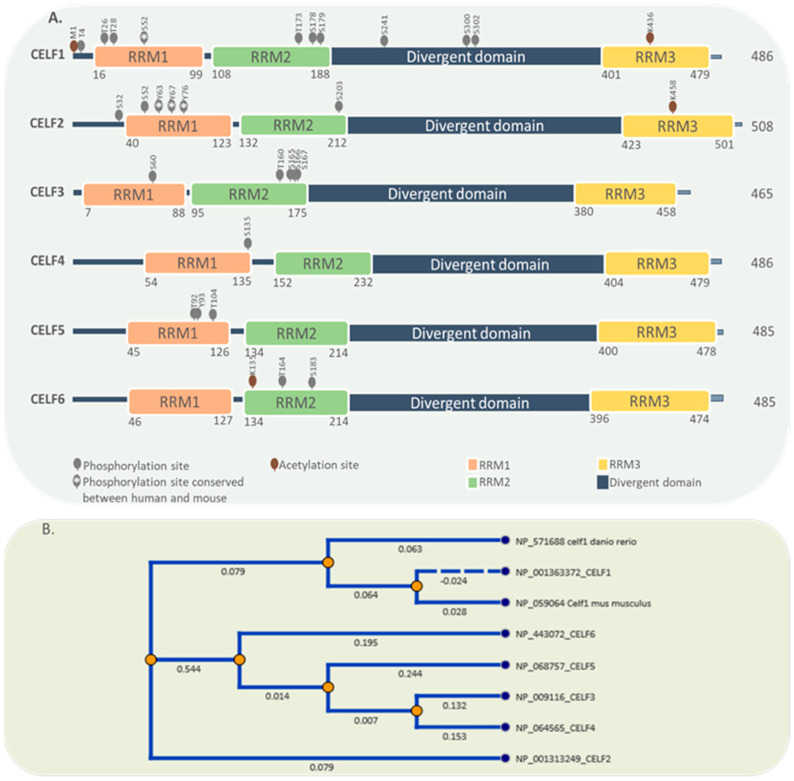
Protein domain structure and post-translational modifications of CELF family members. (**A**) CELF family members share the same domain structure comprising 3 RRMs and a divergent domain. Phosphorylation and acetylation sites are also shown. (**B**) Comparisons of the protein sequences suggest three clear branches for the evolution of CELF members. One branch for CELF1, other for CELF2 and a third for the group 3 to 6.

**Figure 2 ijms-22-11056-f002:**
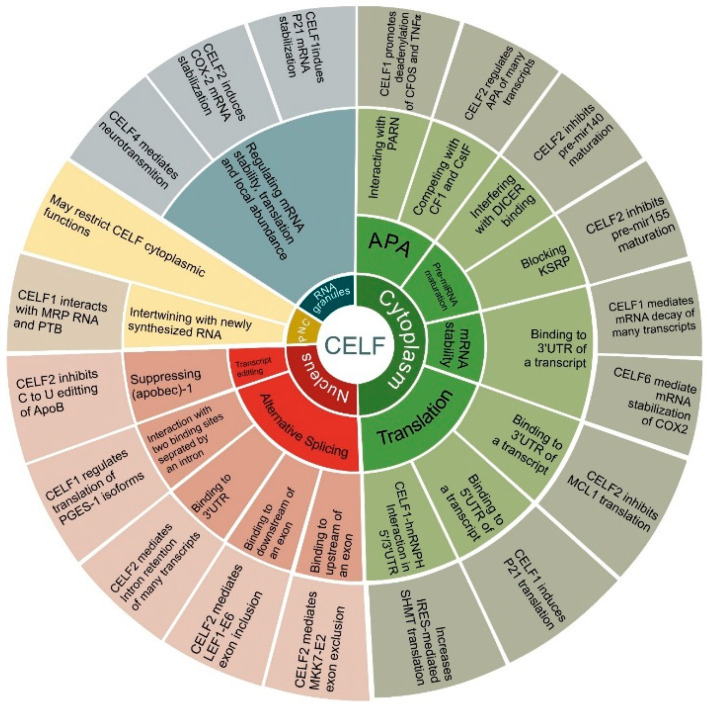
Cellular functions of CELF family members. The circular diagram is divided into 4 main branches illustrating CELF subcellular locations, including the cytoplasm, the nucleus, RNA granules and the peri-nuclear compartment (PNC). Next, each branch is divided into CELFs’ main functions, including translation, mRNA stability, pre-miRNA maturation and alternative polyadenylation (APA) in the cytoplasm, alternative splicing and transcript editing in the nucleus, regulating mRNA stability, translation, local abundance in RNA granules, and, finally, intertwining with newly synthesized RNA in PNCs. In each case, the main mechanism and some representative instances are included for each function. These are discussed in depth in the text.

**Figure 3 ijms-22-11056-f003:**
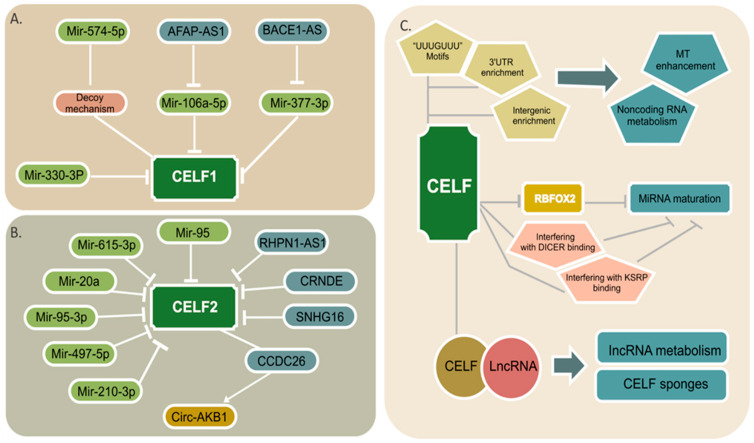
A schematic view of the noncoding RNA/CELF regulatory axis. Multiple miRNAs and lncRNAs targeting each of the CELF1 (**A**) and CELF2 (**B**) proteins are shown. The light green color is representative of miRNAs and the blue color is for lncRNAs. The only circular RNA is shown in mustard color. (**C**) CELF participation in miRNA targeting (MT), noncoding RNA metabolism, and miRNA maturation via both probable and well stablished paths are illustrated. CELF-lncRNA aggregates that are shown in attached circles might play a role in lncRNA metabolism or act as CELF sponges.

**Table 1 ijms-22-11056-t001:** Therapeutic drugs targeting CELF-regulated miRNAs.

miRNA	Molecule/Drug	Effect on miRNA Expression	Cancer Cell Line
mir-20b	5-Fluorouracil	Downregulated	Breast cancer cells [163]
	5-aza-2′-deoxycytidine	Upregulated	Human pancreatic cancer cell lines [164]
miR-106a	Cisplatin	Downregulated	Ovarian cancer cell lines [165]
	Suberoylanilide hydroxamic acid	Downregulated	Human lung carcinoma cell line [166]
	5-Fluorouracil	Downregulated	Colon cancer cell line [167]
	Bicalutamide	Downregulated	Prostate cancer cell line [168]
miR-107	5-Fluorouracil	Upregulated	Breast cancer cells [163]
	Anthranilamide-pyrazolo [1,5-a] pyrimidine	Upregulated	Neuroblastoma cell [169]
miR-140	5-Fluorouracil	Upregulated	Breast cancer cells [163]
miR-140-5p	Suberoylanilide hydroxamic acid	Upregulated	Human lung carcinoma cell line [166]
mir-155	Suberoylanilide hydroxamic acid	Downregulated	Human lung carcinoma cell line [166]
	Ginsenoside Rh2	Downregulated	Human glioma cells [170]
miR-195	5-Fluorouracil	Downregulated	Breast cancer cells [163]
	Trastuzumab	Upregulated	Breast cancer cells [171]
miR-210	5-Fluorouracil	Upregulated	Breast cancer cell line [163]
	Vincristine	Upregulated	Human laryngeal cancer Hep-2 cells [172]
	Ginsenoside Rh2	Downregulated	Human glioma cells [170]
	5-aza-2′-deoxycytidine	Downregulated	Human breast cancer cell line [173]
miR-330	Gemcitabine	Downregulated	Ovarian cancer cell lines [165]
miR-375	5-Fluorouracil	Downregulated	Breast cancer cells [163]
	Suberoylanilide hydroxamic acid	Upregulated	Non-small cell lung cancer cell line [174]
	Trichostatin A	Downregulated	Non-small cell lung cancer cell line [174]
miR-497	5-Fluorouracil	Diferentially expressed on breast cancer cells in treatment group and not in control group	Breast cancer cells [163]
	Bufalin	Upregulated	Human colorectal cancer cell line [175]
miR-574-3p	Suberoylanilide hydroxamic acid (SAHA)	Upregulated	Human lung carcinoma cell line [166]
miR-574-5p	5-Fluorouracil	Upregulated	Breast cancer cells [163]
	Ginsenoside Rh2	Upregulated	Non-small-cell lung cancer cell line [176]
miR-615-3p	Mistletoe lectin-I	Downregulated	CRC cell line CLY (established from liver metastases of a CRC patient) [177]

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
