# Peer review of "CELF Family Proteins in Cancer: Highlights on the RNA-Binding Protein/Noncoding RNA Regulatory Axis"

_ijms, 2021, doi:10.3390/ijms222011056_

Round 1

Reviewer 1 Report

This is a timely, interesting, and very well written review on the CELF family of RNA-binding proteins, with a particular emphasis on their involvement in cancer. I would like to suggest some minor amendments which I feel could improve the manuscript.

  1. I do find the title to be a little bit awkward - why CELF balancing, for example? I would suggest a simpler title.
  2. It would be useful to include a schematic diagram of the structure of CELF family members - RRMs and divergent linkers. Perhaps in section 2.
  3. I also think that the narrative could benefit by some commentary on the evolutionary conservation of CELF family members.
  4. Section 1, line 52 abruptly switches from CELF proteins to ncRNAs, and then describes a "regulatory axis" - I think this could really be explained better, and I would recommend that section 1 is improved with a better flow, and some text clearly explaining the link between CELF and ncRNAs. This is of course explained in detail in later sections, but the link should be established here.
  5. I really admire figure 1; it is quite dense, but clearly outlines the complexity of CELF function. To put this in a broader context, there are of course many examples of multifunctional RBPs so I just wonder if in this section 2, some reference could be made to this - ie, CELF proteins are not unique in being so multifunctional.
  6. CELF1, in terms of its phosphorylation -  is anything known about which splice factor kinases might be involved in this? are other CELFs modified by phosphorylation? what specific residues are phosphorylated? perhaps some additional detail would be beneficial here because splice factor kinases are very clearly implicated in cancer themselves.
  7. Line 379, what are the "CELF6 variants"? please explain. A more general point, the complexity of splice factor function is itself complicated by their own alternative splicing! is there evidence of functionally distinct CELF isoforms (not just CELF6)?
  8. The concluding remarks and future perspectives section is perhaps a little bit long; and some of the material could perhaps fit better in preceding sections.

Author Response

Dear reviewer.

We appreciate the opportunity of improving the manuscript IJMS-1401658 by Nasiri Aghdam. All comments were taken into account. Please see the attachment. 

Best regards, 

Luis Felipe Jave Suárez, Corresponding author

Centro Universitario de Ciencias de la Salud, Instituto de Investigación en Ciencias Biomédicas, Universidad de Guadalajara

Reviewer 2 Report

Thank you for giving me the opportunity of reviewing the manuscript. The authors summarized the previous reports which suggested that CELF (CUGBP Elav like Family) proteins had various functions as RBPs(RNA binding proteins) with a focus on CELF-noncoding RNA in cancer physiology. In this review, the previous reports regarding the diverse mechanisms of CELF proteins as RBPs and CELF-noncoding RNA axis are comprehensively summarized and well organized based on the each function of CELF proteins family and CELF-target noncoding RNA interaction.

As the authors described in the manuscript, CELF and CELF-noncoding RNA axis might be considered as the potential therapeutic targets in cancer therapy. Are there research and reports regarding drug discovery targeting CELF proteins, target noncoding RNA and CELF-noncoding RNA axis? 

Author Response

Dear reviewer.

We appreciate the opportunity of improving the manuscript IJMS-1401658 by Nasiri Aghdam. All comments were taken into account. Please see the attachment. 

Best regards, 

Luis Felipe Jave Suárez, Corresponding author

Centro Universitario de Ciencias de la Salud, Instituto de Investigación en Ciencias Biomédicas, Universidad de Guadalajara, 
